# Physical Validation of a Residual Impedance Rejection Method during Ultra-Low Frequency Bio-Impedance Spectral Measurements

**DOI:** 10.3390/s20174686

**Published:** 2020-08-19

**Authors:** Zoltan Vizvari, Nina Gyorfi, Akos Odry, Zoltan Sari, Mihaly Klincsik, Marin Gergics, Levente Kovacs, Anita Kovacs, Jozsef Pal, Zoltan Karadi, Peter Odry, Attila Toth

**Affiliations:** 1Department of Environmental Engineering, Faculty of Engineering and Information Technology, University of Pecs, Boszorkany str. 2, H-7624 Pecs, Hungary; gyorfi.nina@pte.hu (N.G.); kovacs.anita@mik.pte.hu (A.K.); 2Institute of Information Technology, University of Dunaujvaros, Tancsics M. str. 1/A, H-2401 Dunaujvaros, Hungary; odrya@uniduna.hu (A.O.); podry@uniduna.hu (P.O.); 3Department of Information Technology, Faculty of Engineering and Information Technology, University of Pecs, Boszorkany str. 2, H-7624 Pecs, Hungary; sari.zoltan@mik.pte.hu; 4Department of Mathematics, Faculty of Engineering and Information Technology, University of Pecs, Boszorkany str. 2, H-7624 Pecs, Hungary; klincsik@mik.pte.hu; 51st Department of Medicine, Clinical Centre, University of Pecs, Ifjusag str. 13, H-7624 Pecs, Hungary; gergics.marin@pte.hu; 6Physiological Controls Research Center, University Research and Innovation Cetner, Obuda University, Becsi str. 96/b, H-1034 Budapest, Hungary; kovacs@uni-obuda.hu; 7Institute of Physiology, Medical School, University of Pecs, Szigeti str. 12, H-7624 Pecs, Hungary; pal.jozsef@pte.hu (J.P.); zoltan.karadi@aok.pte.hu (Z.K.); attila.toth@aok.pte.hu (A.T.)

**Keywords:** electrical impedance spectrum measurement, non-invasive testing of biological structures, rejecting measurement errors, residual impedances

## Abstract

Accurate and reliable measurement of the electrical impedance spectrum is an essential requirement in order to draw relevant conclusions in many fields and a variety of applications; in particular, for biological processes. Even in the state-of-the-art methods developed for this purpose, the accuracy and efficacy of impedance measurements are reduced in biological systems, due to the regular occurrence of parameters causing measurement errors such as residual impedance, parasitic capacitance, generator anomalies, and so on. Recent observations have reported the necessity of decreasing such inaccuracies whenever measurements are performed in the ultra-low frequency range, as the above-mentioned errors are almost entirely absent in such cases. The current research work proposes a method which can reject the anomalies listed above when measuring in the ultra-low frequency range, facilitating data collection at the same time. To demonstrate our hypothesis, originating from the consideration of the determinant role of the measuring frequency, a physical model is proposed to examine the effectiveness of our method by measuring across the commonly used vs. ultra-low frequency ranges. Validation measurements reflect that the range of frequencies and the accuracy is much greater than in state-of-the-art methods. Using the proposed new impedance examination technique, biological system characterization can be carried out more accurately.

## 1. Introduction

Electrical impedance spectrum (EIS) measurement is a non-invasive material testing technique. It has been extensively used and applied in several research fields, in order to characterise the electrical properties and study the structures and physico-chemical characteristics of materials. In biological applications, where the goal is the investigation of biological structures, the name of the measurement method is changed to bio-impedance spectrum (BIS) measurement. It has been proven that BIS is an effective technique for non-invasive analyses in several healthy subjects [1,2] and in pathological [3,4,5,6] applications. Furthermore, BIS appears to be an appropriate method for material analysis, not only for human and animal tissue characterisation but also for bacteria ([7,8,9]) and viruses [10,11].

Despite all of the opportunities provided by bio-impedance measurements for healthcare applications, there are still many challenges and open study fields for future research. The problems of sensitivity and precision are the major challenges in these types of applications. Hence, in the case of BIS methods, it is imperative to conduct more extensive investigations. Given that extremely complex biological processes are being monitored, conclusions need to be inferred based on extremely small signal variations. Thus, the improvement of sensitivity has been highlighted in various fields of research [10]. Although BIS techniques are well-known methods, major research efforts need to be made to improving the sensitivity, specificity, and confidence in the use of these techniques [10]. These improvement areas arise from multiple sources of noise and artefacts (e.g., parasitic stray capacitance, impedance mismatch, cross-talk, or their combinations) which can affect the bio-impedance measurement data [12]. The measured data always contain information originating from the measured structure or the actual mixed measurement setup.

The main decisive issue for BIS measurement is the problem of residual impedances, which can influence the accuracy of collected data and lead to imprecise measurements [12]. Based on the study of Fu and Freeborn, the frequency range of 10–100 kHz [12] has been considered the most suitable for BIS measurements [13,14]. Fu and Freeborn observed that larger residual impedances significantly limit the frequency band, resulting in more inaccurate measurements [12]. In the case of reactance, the measured deviations appear at even lower frequencies (≤100 Hz). They appear in all configurations and cause current leakages during measurements. According to the models proposed by Fu and Freeborn, parasitic capacitance results in a decrease in the overall impedance magnitude as a function of frequency. This decrease occurs at lower frequencies for higher values of the parasitic capacitance and higher values of the series residual impedance [12]. Other studies [15,16] have performed the BIS measurements in the wider range of 100 Hz to 100 kHz.

This makes it significantly more difficult to test biological systems, as the data can only be corrected “offline”, after measurements have been carried out by special mathematical procedures, using errors from different measurements. In our article, we aim to solve this problem with the help of a new method that can detect parasitic capacities, residual impedances, generator anomalies, and so on, in parallel with data collection [17]. In the course of data collection, the system’s ground point is connected by the biological sample, where the data are collected in relation to the ground point of the system (i.e., potential measurement). During the data processing step, which is specially designed for this measurement layout, errors can be almost completely eliminated. In contrast to the current solutions, in order to clarify the efficiency and accuracy of the method, we created a validation model to demonstrate the benefits of our method.

The complex impedance of biological structures, examined in a wide frequency range, provides information on the chemical, biochemical, and indirect biological characteristics, in addition to the physical properties of the structure under consideration. The resistivity and permittivity of biological samples are strongly dependent on frequency. Based on theoretical considerations, these phenomena are caused by three main dielectric dispersions. Tang et al. described these dispersions, as follows:The α dispersion that results from the relaxation of ions surrounding the charged cellular membrane appears at low frequencies (from 0 Hz to approximately a few hundred Hertz) [18];At higher frequency ranges (from a few hundred Hz to a few hundred MHz), the β dispersion can be measured. In this frequency range, the cell membrane blocks the ionic flows of intra- and extra-cellular dielectrics by interfacial polarisation (Maxwell-Wagner effect) [18];In the GHz region, the relaxation of free water molecules in the intra- and extra-cellular fluid causes the phenomenon of γ-dispersion [18].

Different mathematical models have been defined to evaluate the BIS characteristics associated with these dispersions. The use of equivalent electrical circuit models creates an opportunity to conduct qualitative analyses in addition to quantitative examinations of biological structures, by fitting to the current measured BIS data [5,19]. The Cole–Cole model, developed by the Cole brothers in 1941, is a relaxation model which is often used to describe dielectric relaxation corresponding to α or β dispersions [20]:(1)$$Z\left(j\omega \right)=R_{\infty }+\frac{R_{0}-R_{\infty }}{1+{\left(j\omega \tau \right)}^{a}},$$
where

Z(jω) is the complex impedance,

$R_{\infty }$ is the resistance corresponding to the *∞* frequency,

$R_{0}$ is the resistance corresponding to 0 Hz frequency,

τ is the time constant,

*a* is the exponent parameter (0<a≤1),

ω is the angular frequency, and

$j=\sqrt{-1}$.

The Cole–Cole model is a non-linear model which uses an electric RC-coupling analogy. The model is based on the replacement of ideal capacitors with more general constant-phase elements (CPEs), owing to the measured electrolyte diffusion features [21]. The *a* exponent parameter (in Equation (Equation 1)) is significant for most tissues, which allows for the description of various spectral shapes. When a=1, it represents the standard Debye model [22]. Therefore, the spectrum of a tissue may be more appropriately described in terms of multiple Cole–Cole dispersions, which can be used in conjunction with a choice of parameters (appropriate to each tissue) to predict the dielectric behaviour over the desired frequency range [17,23].

In theory, the application of the Cole–Cole model cannot be considered a significant challenge, given that this implies fitting the function (Equation (Equation 1)) to data recorded during BIS measurements. However, in practice, this is a non-linear numerical problem for which we need to implement a large amplitude data range. The inferences, based on the literature, are ambivalent with respect to the measurement evaluation methods. There have been many remarkable technological solutions for the extraction of the Cole–Cole parameters. The simplest empirical techniques have been offered in support of the problem regarding several areas based on the use of the Wien bridge oscillator [24,25]. The Cole–Cole data can also be evaluated by introducing various algorithms, such as the Flower Pollination Algorithm (FPA) and the Moth–Flame Optimizer (MFO), among others [26]). Furthermore, based on the relationships among different electrode positions and alterations of the parameters, Freeborn et al. concluded that electrode locations have a significant impact on the extracted parameters [27]. Chen et al. applied a few stochastic and deterministic solvers to compare the median values calculated from Markov chain, Monte Carlo, and Gauss–Newton methods [28]. The Bayesian approach is one of the numerous stochastic methods that has also been used in an attempt to extract the Cole–Cole parameters [29]. Furthermore, a program package was developed and introduced in 2014 for the evaluation of mathematical models regarding different EIS measurements [21].

After studying the relevant literature, we found that the realisation of bio-impedance measurements has primarily focused on β dispersion [10,18]. This implies that the frequency range of the measurements mainly spans between a few tens of hundred Hz and few hundred MHz values. Thus, the detection of α dispersion is not possible, in most cases. In this work, our aim is to introduce a new technology which is capable of measuring bio-impedance precisely by excluding or rejecting the errors and artefacts which occur during data recording, even for frequencies near 0 Hz. Therefore, we sought accurate detection performance, even at extra low frequencies (≤100 Hz).

To demonstrate this, we created a physical model which illustrates the measurement problems described above and their simultaneous rejection. For additional processing of high-purity and -reliability spectral data which are generated during data gathering and processing (i.e., for the extraction of the Cole–Cole parameters), a basic non-linear optimisation process (Levenberg–Marquardt method implemented in MATLAB) is sufficient. The following case study was conducted for a detailed investigation of BIS tests performed on a physical model (BIS phantom).

## 2. Materials and Methods

The most basic methods for realizing BIS methods are two-electrode and four-electrode techniques [30,31,32]. In both of these methods, one of the electrodes is called the driving electrode, through which the current signal is injected, and the other electrodes, through which the frequency-dependent potential is measured, are called sensing electrodes. The two-electrode method uses only two electrodes for BIS measurement and, hence, the current signal injection and voltage measurement are conducted with the same electrodes. As shown in Figure 1a, excitation and measurement are performed at points 1 and 4 and, so, the ratio of the measured voltage ($V_{1,4}$) and the excitation current gives the resultant impedance, which includes the contact impedance.

Therefore, the most significant problem of two-electrode techniques is that the appearance of contact impedances significantly negatively affects the measurement results [30,31]. In contrast, four-electrode techniques are able to ignore the problem of contact impedance [31]. Figure 1a shows that the ratio of voltage between points 2 and 3 ($V_{2,3}$) and the excitation current gives only the $Z_{body}$ impedance. In the case of the implementation of four-electrode techniques, the realization of differential measurements results in significant errors in the data, while the above techniques can only be used by application of a current generator [12]. Therefore, we performed BIS measurements using a method based on the four-electrode technique published by Vizvari et al. [33,34], which is a voltage comparison technique that can be applied to both current or voltage with generator excitation. The authors have shown that the application of this self-developed data acquisition and evaluation process, compared with other techniques, increases the effectiveness and the accuracy of BIS measurements over a wide frequency range [34]. The effectiveness of this method relies on the fact that we implement a common-mode rejection of errors in data collection and data processing specially developed for this purpose [34]. In this case, we wanted to model a life-like measurement scenario (i.e., to maximise the error effects that occur during the measurement). The main principle is illustrated in Figure 1b, where $Z_{in}$ and $Z_{out}$ symbolise the contact impedance corresponding to the electrode and measured object. Of course, the contact impedances do not appear in the $Z_{body}$ impedance signal as a result of the four-electrode measurements, but the measurement errors produced negatively effect the $Z_{body}$ data. As a result of the model’s behaviour at low frequencies, the values of $Z_{in}$ and $Z_{out}$ suppress the $Z_{body}$ impedance. Hence, it does not appear in the resultant impedance of the model (*Z*). Additionally, due to the increased resultant impedance values measured at frequencies near to 0 Hz, the effect of parasite impedances increases. This appears in the measured data, in the resultant impedance (*Z*) spectrum.

The aim of the measurement is to determine the $Z_{body}$ impedance values at many frequency points. The effectiveness of the method relies on the application of $R_{ref}$, a reference resistor, against which the unknown impedance ($Z_{body}$) is compared. As Figure 1b shows, for excitation of the measured object, we use a voltage generator to measure the voltages $u_{1}$, $u_{2}$, $u_{3}$, and $u_{4}$ at nodes 1, 2, 3, and 4, respectively. The precision and efficiency of the method derives from the simple consideration of the digital subtraction of the potentials measured at points 1, 2, 3, and 4 from each other during processing, thus suppressing the various errors that appear in the measured data.

### 2.1. The BIS System

The BIS measuring instrument is a self-developed digital lock-in amplifier built with two 32-bit microcontrollers from STMicroelectronics (hereinafter referred to as MCU1 and MCU2). Three main measurement modes are supported by the instrument. These include signal Fast Fourier transform (FFT) spectral measurement, frequency sweeping impedance measurement (EIS), and impedance measurements at fixed frequencies. In the FFT spectral measurements, MCU1 transmits the desired excitation signal (which is set up in the GUI) to the external DAC, while MCU2 collects the raw ADC measurements, executes the FFT algorithm, and sends the spectral results back to MCU1. In the case of EIS (which is used for BIS measurements), the user is required to select the decades of interest (frequency range) and the number of measurement frequencies in each decade. Then, MCU1 calculates the sinusoidal signal to be swept through the selected frequency decades and performs signal excitation. MCU2 performs the digital lock-in amplifier algorithm for each selected frequency during the sweeping procedure and calculates the instantaneous amplitude and phase results. In case of impedance measurement at fixed frequencies, MCU1 performs excitation at the desired frequency, while MCU2 calculates the impedance results that correspond to this frequency.

In addition to the advantages mentioned so far, the self-developed measuring system naturally has limitations, the most important of which are listed below:The highest processable frequency of the reference signal, according to the calculation capacity of the controller used in the instrument, is 100 kHz;in the case of in-situ measurements, there are measurement time limits which cannot be reduced, due to the characteristics of the measurement principle applied;the excitation current must be kept below 1 mA and the excitation tension may be up to 10 V;it is Battery-powered, allowing us to use the instrument for at least 6.5 h in autonomous operation.

MCU1 is the master unit and is programmed to:receive the measurement commands through a universal asynchronous receiver–transmitter (UART1) peripheral sent by the personal computer (PC) user with the help of the graphical user interface (GUI);both set up and establish the $I^{2}S$ communication [35] with the external digital-to-analog converter (DAC);calculate and transmit the excitation signal to the DAC; andreceive the measurement results of MCU2 by UART2 and send these back to the GUI by UART1.

MCU2 works as the slave unit and is responsible for:setting up its $I^{2}S$ peripheral and establishing communications for receiving the real-time measurements of the external analog-to-digital converter (ADC) on four channels;executing the digital lock-in amplifier algorithm based on the excitation signal and measurement results, and calculatinge the real-time amplitude and phase results corresponding to the excitation frequency; andsending the instantaneous impedance results back to the master unit (MCU1).

Figure 2a shows the working principle of the equipment in a block-diagram form, as well as the realized equipment.

Our research group developed the digital lock-in algorithms [36], following the improvement of the previous software published by Vizvari et al. [34], and their optimisations for this four-channel instrument (in particular, with respect to the measurement noise). The lock-in measurement procedure around the reference frequency selects signals in a specific frequency range, effectively discarding all other frequency components from the signal, thus having a selective filter character. Lock-in amplifiers are able to measure the amplitude and phase of the signal relative to a specific reference signal, even if the signal is entirely noise. By applying the work described in the documentation of Zurich Instruments [37], we can review the basics of lock-in amplifier operation. Using the instructions in [38], we estimated the noise characteristics of our instrument and the lock-in amplifier gain during the measurement process, as well as deriving numerical measurements of the output signal-to-noise ratio relative to the input signals, thus demonstrating the ways in which the lock-in amplifier can improve its signal-to-noise ratio during measurement.

By analysing the operation of the lock-in amplifier, it can be assumed that the noise was not correlated with the signal $V_{s}\left(t\right)$, such that they can be analysed separately [38]. First, we analyse the transfer of $V_{s}\left(t\right)$ signal through the lock-in amplifier:
(2a)$$V_{r}\left(t\right)=V_{r}\cdot cos\left(2\pi f_{r}t-\varphi_{r}\right),$$
(2b)$$V_{s}\left(t\right)=V_{s}\cdot cos\left(2\pi f_{s}t-\varphi_{s}\right),$$
where

$V_{r}$ is the amplitude of the reference signal,

$\varphi_{r}$ is the phase of the reference signal,

$f_{r}$ is the frequency of the reference signal,

$V_{s}$ is the amplitude of the measured signal,

$\varphi_{s}$ is the phase of the measured signal, and

$f_{s}$ is the frequency of the measured signal.

According to the phase sensitive lock-in principle, the measured signal is multiplied by cosine and sine reference signals (Figure 3). Applying the equality $f_{r}=f_{s}=f$ gives
(3a)$$V_{A}\left(t\right)=V_{s}\cdot cos\left(2\pi ft-\varphi_{s}\right)\cdot V_{r}\cdot cos\left(2\pi ft-\varphi_{r}\right),$$
(3b)$$V_{B}\left(t\right)=V_{s}\cdot cos\left(2\pi ft-\varphi_{s}\right)\cdot V_{r}\cdot sin\left(2\pi ft-\varphi_{r}\right).$$

By rearranging Equation (3a,b), we get the following equations:
(4a)$$V_{A}\left(t\right)=0.5\cdot V_{s}\cdot V_{r}\cdot \left[cos\left(2\cdot 2\pi ft-\varphi_{s}-\varphi_{r}\right)+cos\left(\varphi_{r}-\varphi_{s}\right)\right],$$
(4b)$$V_{B}\left(t\right)=0.5\cdot V_{s}\cdot V_{r}\cdot \left[sin\left(2\cdot 2\pi ft-\varphi_{s}-\varphi_{r}\right)+sin\left(\varphi_{r}-\varphi_{s}\right)\right].$$

Application of a low-pass filter with attenuation A(2·2πft) at frequency 2·2πft gives
(5a)$$V_{X}\left(t\right)=0.5\cdot V_{s}\cdot V_{r}\cdot A\left(2\cdot 2\pi ft\right)\cdot cos\left(2\cdot 2\pi ft-\varphi_{s}-\varphi_{r}\right)+0.5\cdot V_{s}\cdot V_{r}\cdot cos\left(\varphi_{r}-\varphi_{s}\right),$$
(5b)$$V_{Y}\left(t\right)=0.5\cdot V_{s}\cdot V_{r}\cdot A\left(2\cdot 2\pi ft\right)\cdot sin\left(2\cdot 2\pi ft-\varphi_{s}-\varphi_{r}\right)+0.5\cdot V_{s}\cdot V_{r}\cdot sin\left(\varphi_{r}-\varphi_{s}\right).$$

Integration of Equation (5a,b) over intervals which are whole-number multiples of sine and cosine periods gives
(6a)$$\bar{V_{X}\left(t\right)}=0.5\cdot V_{s}\cdot V_{r}\cdot cos\left(\varphi_{r}-\varphi_{s}\right),$$
(6b)$$\bar{V_{Y}\left(t\right)}=0.5\cdot V_{s}\cdot V_{r}\cdot sin\left(\varphi_{r}-\varphi_{s}\right).$$

If integration is effectively implemented, the low-pass filter has no effect on signal processing. The integration (applied in Equation (4a,b)) produces the same result (Equation (6a,b)), even without use of a low-pass filter.

To study the effect of noise, we define the noise components obtained at points A and B in Figure 3, which are obtained after multiplying:
(7a)$$V_{nA}\left(t\right)=n\left(t\right)\cdot V_{r}\cdot cos\left(2\pi f_{r}t-\varphi_{r}\right),$$
(7b)$$V_{nB}\left(t\right)=n\left(t\right)\cdot V_{r}\cdot sin\left(2\pi f_{r}t-\varphi_{r}\right).$$

For both of the $V_{nA}\left(t\right)$ and $V_{nB}\left(t\right)$ components, the incoming noise spectrum is shifted in the frequency (modulated) by the reference signal in the frequency band. Therefore, we receive two-sided band modulated noise (Figure 4) around the reference frequency (therefore, the noise with 1/f characteristics is removed in the low-frequency range and the filter can cut it out better). Depending on its slope, the low-pass filter cuts out the low-frequency part of the noise and the noise component remaining after filtering loses its white-noise properties [37].

Band-pass and low-pass filters are optional; as such, they form a reserve of the measuring system. In Figure 3, “band-pass filter 2” is responsible for setting the same signal delay as the “band-pass filter 1”. In the case of our self-developed bioimpedance measurement techniques, the range of system dynamics has proved sufficient so far. Thus, low-pass and band-pass filters have not yet been used in our bioimpedance measurements. This is beneficial for us, allowing us to avoid longer setup (and therefore measurement) times; however, some applications or testing options may allow us to take advantage of filters. In this case, two filters are applied:for 37.5 ksample/s, a first-degree and second-degree filter with cut-off frequency of 0.05 Hz in the frequency ranges 0.01 Hz and 10 kHz, respectively; andfor 375 ksample/s, a filter with cut-off frequency of 0.5 Hz is applied from 10 kHz to 100 kHz.

If measurements are made between 1 mHz and 0.01 Hz, the low-pass filter is tuned separately. In this work, the low-pass filter was not applied.

The above-detailed changes comprise a significant improvement, compared to the instruments published by Vizvari et al. [34]. Of course, the further development of the measuring system has been carried out in other ways. The comparison between the work published by Vizvari et al. [34] and the improved four-channel instrument is detailed as follows:-The new instrument is based on 32-bit ARM microcontrollers running on 200 MHz clock frequencies, instead of an FPGA circuit;-This instrument employs 32-bit ADC and DAC circuits while, in the previous work, 24-bit data converters were used. Moreover, the analog circuits in the generator are characterized by reduced total noise in the improved four channel instrument;-The inputs of measurement circuits are characterized by 3.2 nV/$\sqrt{Hz}$ and 200 fA/$\sqrt{Hz}$ input noises at 1 kHz (the measurement load generates much bigger noise). Additionally, the amplification factor of the input circuit is 1;-The ARM microcontrollers in the four-channel instrument execute the algorithm in the IEEE.754 64-bit double precision domain, while the FPGA in the previous work employed 48-bit fixed point arithmetic for the digital lock-in algorithm; and-In the case of sweeping impedance measurement in the new instrument, the frequency change is realized at zero crossing points in the generator, in order to increase the measurement speed (and decrease the settling time). Moreover, there is an option to set the frequency change at the maximum point of the sine wave; however, this option was not used in the proposed measurement.

The proposed measurement method is characterized by the following properties:-During the sweeping measurement process, two sampling frequencies (fs) are employed. In the 10–100 kHz range, $f_{s}=375$ ksample/s is used; while, in the 1 mHz–10 kHz frequency range, the signals are sampled at $f_{s}=37.5$ ksample/s. If $f_{s}=375$ ksample/s sampling is executed, then the data handling for real-time calculations is realized with ping-pong buffering, as discussed in [35];-In each decade during the sweeping measurement, the number of excitation frequencies can be chosen between 3 and 100 (the frequency values are selected equidistantly on the logarithmic scale);-A first-order complementary filter-based band-pass filter smooths the raw ADC signals before the execution of the lock-in algorithm (this feature can be turned on and off in the GUI). In the presented measurements, these filters were turned off;-The executed algorithm contains low-pass filters after the signal multiplications, which can be also turned on and off in the GUI. These filters are first- or second-order RC circuit-based filter structures; and-The sweeping impedance measurement is executed with different integration times in the case of different frequencies. The durations of measurements, in different frequency decades (in the case of 33 frequency points per domain), are:
132 s in the 100 kHz–10 Hz range40 s in the 10 Hz–1 Hz range180 s in the 1 Hz–0.1 Hz range1800 s in the 0.1 Hz–0.01 Hz rangenear to 18,000 s in the 0.01 Hz–1 mHz range.

The aforementioned ranges were selected empirically (i.e., these integration times are the multiples of the period of the sine wave excitation signal). The properties of the realized complete measuring system (Figure 2d) are as follows:-the excitation signal is a monochromatic sine wave in the frequency range between 1 mHz and 100 kHz, with a total harmonic distortion and noise (THD+N) greater than 100 dB;-the excitation is provided by a voltage generator with a maximum noise level of 1.5 $\mu V_{eff}$ in the frequency range of 1 mHz to 100 kHz;-the range of excitation is 110 dB for voltage-generator modes, with a maximum value of 10 V peak-to-peak; and-the accuracy of the measured data is less than 1 part per million (ppm) for amplitudes and for phase less than 0.01${}^{\circ }$.

### 2.2. BIS Measurement Setup

To validate the self-developed BIS system and the data acquisition process, we constructed a BIS phantom (Figure 5b), based on the publication of Fu and Freeman [12]. Our main goal in designing the phantom was to measure the α-dispersion with high accuracy at ultra-low frequencies. The phantom (shown in Figure 5a) is a passive electric circuit which produces a typical ultra-low frequency bio-impedance spectrum.

The phantom (shown in Figure 5a) was built using commercially available resistors ($R_{in}=R_{out}$ = 100 kΩ, $R_{\infty }$ = 1 kΩ and $R_{0}-R_{\infty }$ = 10 kΩ) and capacitors ($C_{in}=C_{out}=100\;\mu$F and C=10μF), with respective resistor and capacitor tolerances of 1% and 5%. These parts allowed us to calculate the theoretical values of impedances, as follows:(8)$$Z_{in}=\frac{1}{\frac{1}{R_{in}}+j\omega C_{in}},$$
(9)$$Z_{body}=R_{\infty }+\frac{1}{\frac{1}{R_{0}-R_{\infty }}+j\omega C},$$
(10)$$Z_{out}=\frac{1}{\frac{1}{R_{out}}+j\omega C_{out}},$$
(11)$$Z=Z_{in}+Z_{body}+Z_{out}.$$

These impedance values could be used for the validation of the BIS equipment by comparing them with their measured counterparts. The $Z_{body}$ impedance (Equation (Equation 9)) is equal to the so-called Debye model, which follows from the Cole–Cole model by setting a=1 in Equation (Equation 1).

The validation of the BIS instrument (described in Section 2.1) was performed based on the construction of a measurement setup based on the BIS phantom shown in Figure 5a. Figure 5a also shows the realised validation setup, which was constructed based on the measurement principle depicted in Figure 1b.

In addition to the use of the components of the BIS phantom (Figure 5a), we built a reference resistor into the measurement circuit: $R_{ref}=100\;\Omega$ (in Figure 5a).

### 2.3. Data Acquisition and Evaluation Methods

As a first step for physical validation, FFT spectra were measured to characterise the measuring circuit. To perform the FFT measurements, we applied a voltage generator with 1 V (peak-to-peak) amplitude and 0.8 Hz frequency. The measured spectra were recorded only in the case of $u_{2}$ and $u_{3}$ signals (based on circuit shown in Figure 5a) in the 10 mHz–100 kHz frequency range. The BIS data acquisition in the setup shown in Figure 5a was performed with the use of 1 V (peak-to-peak) excitation signals in a wide frequency range of 1 mHz to 100 kHz. To prove the consistency and effectiveness of the instrument and measuring method, we performed three types of spectral recordings:Five data points per decade (40-point spectrum),Ten data points per decade (80-point spectrum), andThirty-three data points per decade (264-point spectrum).

All measurements were repeated five times. In the case of the validation setup (Figure 5a), the measured voltage data was evaluated by the following calculations [34], based on the measurement principle published by Vizvari et al. [34] and described by Figure 1b:(12)$$Z_{in}^{\prime }=R_{ref}\cdot \frac{u_{1}-u_{2}}{u_{4}},$$
(13)$$Z_{body}^{\prime }=R_{ref}\cdot \frac{u_{2}-u_{3}}{u_{4}},$$
(14)$$Z_{out}^{\prime }=R_{ref}\cdot \frac{u_{3}}{u_{4}}-1,$$
(15)$$Z^{\prime }=R_{ref}\cdot \frac{u_{1}}{u_{4}}-1,$$
where $Z_{in}^{\prime }$, $Z_{body}^{\prime }$, $Z_{out}^{\prime }$, and $Z^{\prime }$, are the measured counterparts of $Z_{in}$ (Equation (Equation 8)), $Z_{body}$ (Equation (Equation 9)), $Z_{out}$ (Equation (Equation 10)), and *Z* (Equation (Equation 11)), respectively. The impedances, calculated by Equations (Equation 8)–(Equation 15) are complex numbers. Thus, we used the amplitude and angle errors for further display in the Bode diagrams.

At this point, we calculated the averages of similarly recorded spectra and performed the first step of validation, based on the calculation of the relative errors of the measured values:(16)$$\epsilon_{in}=100\cdot \frac{|Z_{in}^{\prime }-Z_{in}|}{|Z_{in}|},$$
(17)$$\epsilon_{body}=100\cdot \frac{|Z_{body}^{\prime }-Z_{body}|}{|Z_{body}|},$$
(18)$$\epsilon_{out}=100\cdot \frac{|Z_{out}^{\prime }-Z_{out}|}{|Z_{out}|},$$
(19)$$\epsilon_{Z}=100\cdot \frac{|Z^{\prime }-Z|}{\left|Z\right|}.$$

The second step of validation was the extraction of the Cole–Cole model parameters and comparison with their theoretical counterparts. To accomplish this, we applied a simple non-linear Levenberg–Marquardt-based regression algorithm in MATLAB. We performed the curve-fitting procedure on each recording and calculated the relative errors of the Cole–Cole model parameters, as follows:(20)$$\epsilon_{a}=100\cdot \frac{|a^{\prime }-a|}{a},$$
(21)$$\epsilon_{\tau }=100\cdot \frac{|\tau^{\prime }-\tau |}{\tau },$$
(22)$$\epsilon_{R_{0}}=100\cdot \frac{|R_{0}^{\prime }-R_{0}|}{R_{0}},$$
(23)$$\epsilon_{R_{\infty }}=100\cdot \frac{|R_{\infty }^{\prime }-R_{\infty }|}{R_{\infty }},$$
where $a^{\prime }$, $\tau^{\prime }$, $R_{0}^{\prime }$, and $R_{\infty }^{\prime }$ are the extracted parameters and *a*, τ, $R_{0}$, and $R_{infty}$ are their respective theoretical counterparts. From Section 2.2, it follows that the theoretical values were a=1, $R_{0}$ = 11 kΩ, and
(24)$$\tau =(R_{0}-R_{\infty })\cdot C=0.1\;s.$$

Using the equations above (Equations (Equation 20)–(Equation 23)), we compared the calculated relative errors with the tolerances of the resistors and capacitors. We examined whether the relative errors of the measurements remained within the set tolerance.

## 3. Results and Discussion

The FFT spectra of $u_{2}$ and $u_{3}$ signals (Figure 6a,b) show the peaks of the measured signals at a very low noise levels. For the whole frequency range, the noise level usually remained below −120 dB. This is one of the most important property that forms the basis of outstanding precision of our technology.

Given that the purpose of BIS measurements is to determine the body’s impedance spectrum ($Z_{body}$) as accurately as possible, the consideration of $Z_{in}$ and $Z_{out}$ impedances was deemed not necessary during further evaluations. Given that the impact of impedance is not negligible for phantom measurements, the purpose of four-electrode measurements is to reject the measurement errors and noise of spectra as efficiently as possible. To illustrate the effectiveness of rejection, we continued to rely on the evaluation of the body ($Z_{body}$) and the resulting impedance (*Z*) spectra.

All of the related materials, including the MATLAB files, the measurements, and models, have been made publicly available in the supplementary online material [39].

The application of the phantom offered us the opportunity to calculate the theoretical values corresponding to the measured frequency points. Hence, in the first step of validation, the average spectra were compared with the theoretical values of the calculated counterpart spectra (as described in Section 2.3). A comparison of the theoretical and measured (40-point) spectra of $Z_{body}$ and *Z* are shown in Figure 7a,b, respectively.

The measured and calculated $Z_{body}$ spectra were in close agreement. Apparently, there were no significant differences between the curves. Based on Equation (Equation 17), the maximum relative error (calculated from the entire set of relative error data) was max($\epsilon_{body}$) = 3.016%. These error values increased when the resultant impedance of the phantom was measured: max($\epsilon_{Z}$) = 10.968%.

The spectra recorded with 80 frequency points gave similar relative error values: max($\epsilon_{body}$) = 2.965% for $Z_{body}$ and max($\epsilon_{Z}$) = 9.909% for *Z*. The comparison of the measured and calculated spectra is shown in Figure 7c,d.

The spectra that consisted of the highest number of frequency points in this validation procedure (i.e., 264 points), had similar relative error values: max($\epsilon_{body}$) = 3.196% for $Z_{body}$ and max($\epsilon_{Z}$) = 9.089% for *Z*. The comparison of the measured and calculated spectra is shown in Figure 7e,f.

For these three recordings (as observed in Figure 7b,d,f), a decrease in the frequency increased the difference between the measured and calculated curves. Based on [12], the impedances $Z_{in}$ and $Z_{out}$ were responsible for the increase of the error rate. This was also reflected in the maximum values of the relative errors of the *Z* curves. The application of the four-electrode arrangement (as can be observed in Figure 7a,c,e, and the corresponding relative error values) rejected this effect and stabilised the spectrum. The frequency-dependent error shown above was significantly reduced in this case.

In the second step of validation, as was described in Section 2.3, the Cole–Cole parameters were extracted for each BIS data set. Table 1 shows the goodness-of-fit (Equation (Equation 25)) and summarises the Cole–Cole parameters calculated from the 40-point spectra.
(25)$$R^{2}=1-\frac{\sum_{i=1}^{n}{\left({\hat{Z}}_{body_{i}}-Z_{body_{i}}^{\prime }\right)}^{2}}{\sum_{i=1}^{n}{\left({\bar{Z}}_{body}^{\prime }-Z_{body_{i}}^{\prime }\right)}^{2}},$$
where

$Z_{body_{i}}^{\prime }$ is the $i^{\mathrm{th}}$ data point in $Z_{body}^{\prime }$,

${\hat{Z}}_{body_{i}}$ is the $i^{\mathrm{th}}$ data point in the BIS data calculated from the extracted Cole–Cole parameters (corresponding to the frequency points of the actual measurement),

${\bar{Z}}_{body}^{\prime }$ is the average of $Z_{body}^{\prime }$ data, and

*n* is the number of data points in $Z_{body}^{\prime }$.

**Table 1 sensors-20-04686-t001:** Extracted Cole–Cole parameters and relative errors (Equations (Equation 20)–(Equation 23)) calculated from the 40-point BIS data.

Meas. No.	$R^{2}$ (-)	*a* (-)	$\epsilon_{a}$ (%)	τ (s)	$\epsilon_{\tau }$ (%)	$R_{0}$ (Ω)	$\epsilon_{R_{0}}$ (%)	$R_{\infty }$ (Ω)	$\epsilon_{R_{\infty }}$ (%)
1.	0.99991	0.9687	3.23	0.1017	1.72	11,107	0.96	996.31	0.37
2.	0.99998	0.9867	1.34	0.0993	0.73	10,941	0.53	1004.2	0.42
3.	0.99998	0.9796	2.07	0.1002	0.19	11,006	0.05	1001.2	0.12
4.	0.99999	0.9845	1.57	0.0996	0.36	10,965	0.32	1003.2	0.32
5.	0.99999	0.9833	1.69	0.0998	0.22	10,976	0.22	1002.7	0.27

In general, there was an extremely good correlation between the measured data and the fitted Cole–Cole model function. In Table 1, the goodness-of-fit can be seen to remain near unity, where the minimum value ($R^{2}=0.99991$) belonged to the first measurement. As Figure 8 shows, even in case of the “worst” fitting, the extracted Cole–Cole parameters accurately represented the parameters of the phantom (first row in Table 1).

The errors $\epsilon_{\tau }$ (Equation (Equation 21)), $\epsilon_{R_{0}}$ (Equation (Equation 22)), and $\epsilon_{R_{\infty }}$ (Equation (Equation 23)) were less than 1% in the case of the 40-point measurement data. The most significant errors appeared in the parameter *a*, but these values were still smaller than the tolerance value of the capacitors (5%).

As a result of the increase in the number of frequency points, no significant changes were observed in the model fitting results (Table 2).

The relative error values still remained less than 1%, except for $\epsilon_{a}$, which was still within the capacitor tolerance ranges. The first row in Table 2 contains the parameters extracted for the “worst” fit ($R^{2}=0.99997$), as depicted by Figure 9.

The 264-point BIS recordings still gave similar results to those yielded by the other data. However, in comparison with the other data (Table 1 and Table 2), Table 3 lists higher relative error values. However, in general, the relative errors were still less than the tolerance values. Thus, the Cole–Cole parameters extracted from the 264-point BIS data were also acceptable.

In a similar manner to the previous cases, we show the “worst”-fitted Cole–Cole model in Figure 10, which shows that the measured impedance values became noisy at frequencies less than 0.1 Hz. However, the fitting process adequately compensated for the noisy data sets.

In summary, the validation procedure can be considered successful. In the first step, the effectiveness and robustness of the data-collection protocol was demonstrated, based on the comparison of the maximum values of the relative errors that corresponded to the two- and four-electrode measurements. Based on this, it can be concluded that contact impedances can be ignored by applying the self-developed data acquisition and evaluation method, resulting in a better rejection of noise, as the relative errors were smaller in the case of four-electrode measurements. Furthermore, by comparing the relative errors of the phantom, it was shown that the errors obtained by the four-electrode measurements strictly remained within the tolerance limits.

In the second step of validation, the Cole–Cole parameters were extracted from the BIS data. With the relative errors of the model parameters, the accuracy of the parameters was demonstrated. Accordingly, it can be concluded that all the BIS data provided exact model parameters with minimal relative errors. Table 4 presents the averages of the extracted Cole–Cole parameters:

Based on the consideration of these results, the validation of the proposed BIS instrument and measurement method was successful. In general, the results remain within the limits of the phantom tolerance. Furthermore, repeating the data collection using several frequency point sets did not affect the accuracy of the measured parameters. This justifies the robustness and effectiveness of the BIS measurement technique.

It is crucial to ensure that the recorded data are free from artefacts and that they can be correctly analysed. Successful analysis requires acquisition of the cleanest and most reliable data. Considering these requirements, a need emerges for the development of a measurement technique that can reject these effects over a wide frequency range. Our research activity concentrated on the lower frequency band, ranging from direct current (DC, 0 Hz) to a few hundred kHz. The main goal in our field of research is the minimisation of the effect of the aforementioned errors at the hardware level. We investigated the phenomena that originate from bio-impedance measurement errors (e.g., parasitic capacitances, residual impedances, generator errors, and so on) in depth. Hence, we developed a robust measurement technique for data collection and pre-processing which is free from such errors, effectively increasing the sensitivity of measurement.

## 4. Conclusion and Future Works

As far as the BIS technique is concerned, the interpretation of biological data is significantly hampered by the residual impedances and other measurement errors that render it completely incomprehensible in many cases. The importance of ultra-low frequency bio-impedance measurements refers to the fact that measurement at 50 kHz is commonly used in full-body measurements, which completely ignores the parameters of the extra-cellular space. Given this background, we developed a system that can provide adequate answers to the questions raised previously. We created an ultra-low frequency measurement procedure which fully displays α dispersion across pure, raw data.

We created a full-body phantom that allowed us to extract data from our previously registered data which are free of residual impedance and other parasitic effects. To confirm our claim, the Cole–Cole model was fitted to the measured spectral data and the parameters of the model were extracted. The statistical analysis indicating the goodness-of-fit (i.e., the correlation coefficient) clearly demonstrated that we recovered the values from the validation measurements almost perfectly. The accuracy and reproducibility of the data measured by the elimination of errors which cause the variability of bio-impedance measurements is expected to significantly increase.

One area of potential use appears to be high-precision full-body measurements [10,17], bio-impedance measurements in different parts of the body [40], and multi-sensory applications, such as non-invasive blood glucose measurements [41]. An additional area of promising application may be represented by measurements on cell lines [14,42]. By complementing this method, current bio-impedance measurements can be obtained from previously undetectable but biologically relevant data. The basic feature of this method is that it can be adapted extremely well to issues other than those listed above. Adapting this technology to specific needs (by, e.g., replacing a reference element or adding new hardware elements) provides an opportunity to adapt it to measurements related to environmental [43,44] and biotechnological [45,46] research.

## Figures and Tables

**Figure 1 sensors-20-04686-f001:**
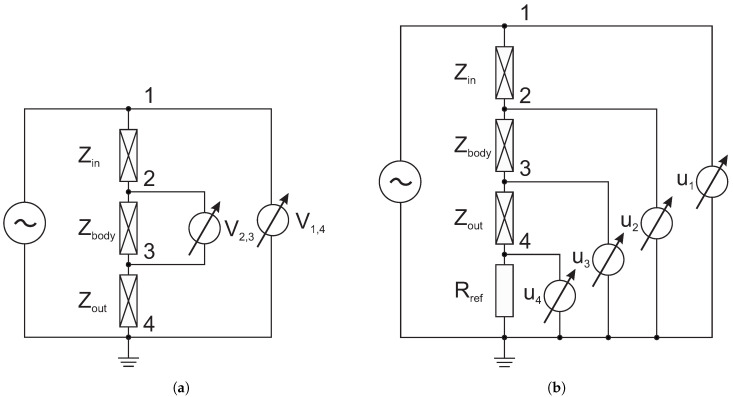
Comparison of the commonly applied techniques and our proposed BIS measurement technique: (**a**) The common BIS measurement principle, where $Z_{in}$ and $Z_{out}$ symbolise the contact impedance corresponding to the electrode and measured object $Z_{body}$; $Z_{body}$ can be calculated using the measured voltages ($V_{1,4}$ in case of two-electrode, or $V_{2,3}$ in case of four-electrode techniques) only for current generator. (**b**) The BIS measurement principle, where $Z_{in}$ and $Z_{out}$ symbolise the contact impedance corresponding to the electrode and measured object $Z_{body}$; $Z_{body}$ can be calculated using the measured potentials ($u_{1}$, $u_{2}$, $u_{3}$, $u_{4}$) for both current or voltage generators.

**Figure 2 sensors-20-04686-f002:**
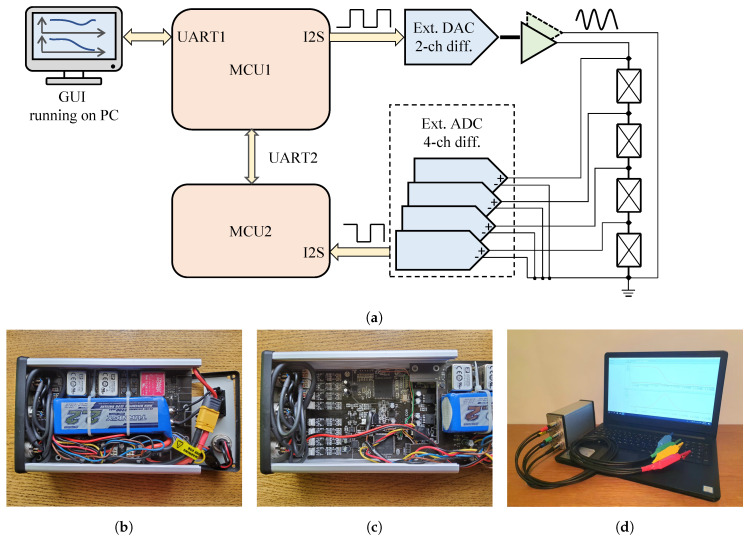
The BIS system (measuring software, measuring device, measuring cable, preamplifiers): (**a**) Block diagram of bio-impedance spectrum (BIS) measuring instrument; (**b**) The realized measuring system (power supply); (**c**) The realized measuring system (measuring board); and (**d**) Complete measuring system for impedance spectrum measurement.

**Figure 3 sensors-20-04686-f003:**
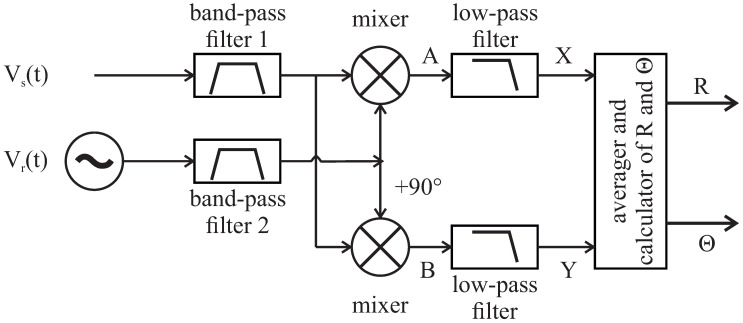
The lock-in principle (where R denotes the amplitude and Θ is the phase of measured signal).

**Figure 4 sensors-20-04686-f004:**
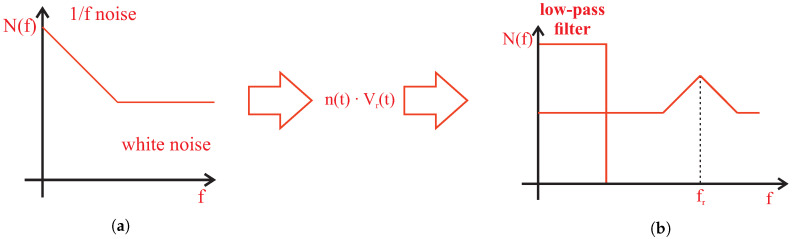
The effect of low-pass filtering, according to the noise in lock-in principle: (**a**) The incoming noise spectrum (baseband noise); and (**b**) The position of the low-pass filter and modulated noise in the spectrum.

**Figure 5 sensors-20-04686-f005:**
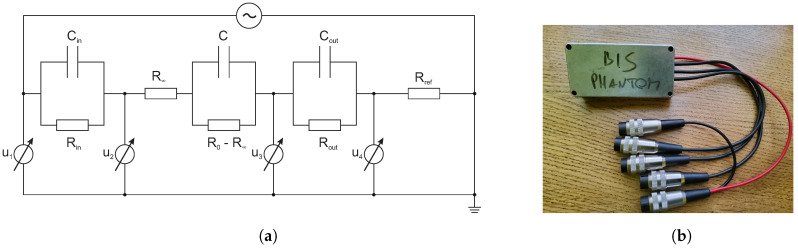
The BIS phantom constructed for physical validation: (**a**) Measurement setup used for the validation of the BIS instrument ($R_{in}=R_{out}$ = 100 kΩ, $R_{\infty }$ = 1 kΩ, $R_{0}-R_{\infty }$ = 10 kΩ, $C_{in}=C_{out}=100\;\mu$F, and C=10μF); and (**b**) The implemented phantom.

**Figure 6 sensors-20-04686-f006:**
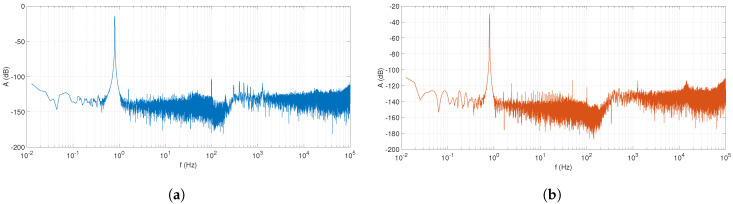
The measured FFT spectra measured by the setup shown in Figure 5a for characterizing the noise levels of the $u_{2}$ and $u_{3}$ signals: (**a**) The FFT spectrum of the $u_{2}$ signal (excitation frequency 0.8 Hz); and (**b**) The FFT spectrum of the $u_{3}$ signal (excitation frequency 0.8 Hz).

**Figure 7 sensors-20-04686-f007:**
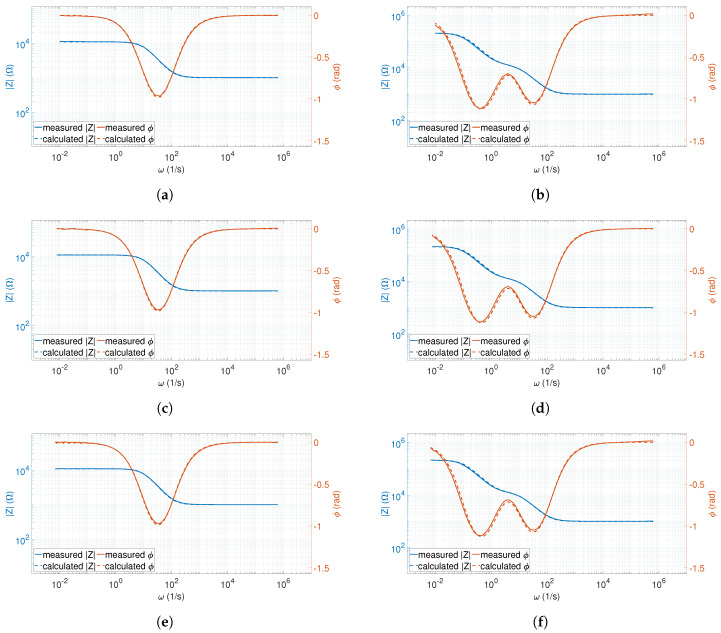
The measured vs. calculated values of $Z_{body}$ and *Z* spectra: (**a**) Measured vs. calculated $Z_{body}$ spectra recorded at 40 points (an example from the 40-point data set); (**b**) Measured vs. calculated *Z* spectra recorded using 40 points (an example from the 40-point data set); (**c**) Measured vs. calculated $Z_{body}$ spectra recorded with 80 points (an example from the 80-point data set); (**d**) Measured vs. calculated *Z* spectra recorded with 80 points (an example from the 80-point data set); (**e**) Measured vs. calculated $Z_{body}$ spectra recorded with 264 points (an example from the 264-point data set); and (**f**) Measured vs. calculated *Z* spectra recorded with 264 points (an example from the 264-point data set).

**Figure 8 sensors-20-04686-f008:**
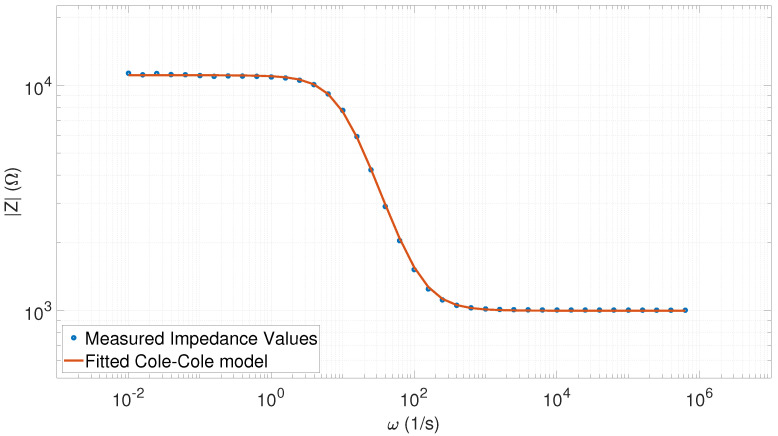
Fitted Cole–Cole model with the measured impedance spectrum values (40 points) corresponding to the $R^{2}=0.99991$ value (a=0.9687, τ=0.1017, $R_{0}$ = 11,107, and $R_{\infty }=996.31$).

**Figure 9 sensors-20-04686-f009:**
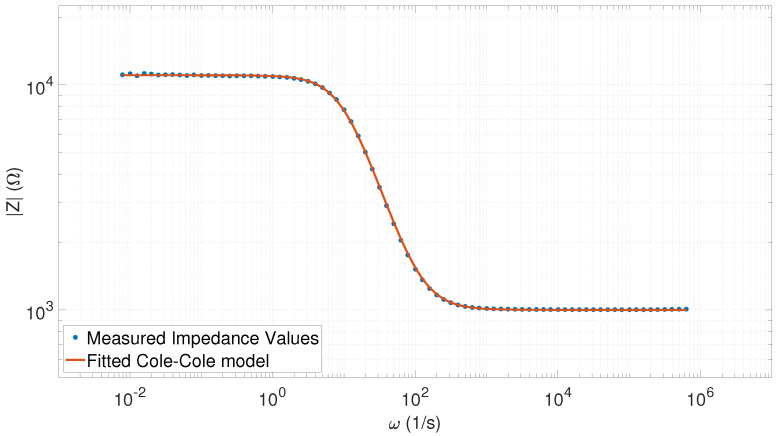
Fitted Cole–Cole model with the measured impedance spectral values (80 points) corresponding to the $R^{2}=0.99997$ value (a=0.9784, τ=0.1005, $R_{0}$ = 11,046, $R_{\infty }=999.22$).

**Figure 10 sensors-20-04686-f010:**
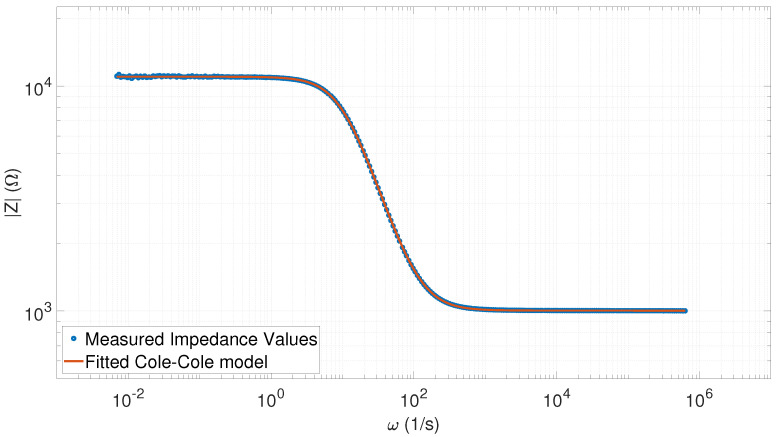
Fitted Cole–Cole model with the measured impedance spectrum values (264 points) corresponding to the fifth measurement ($R^{2}=0.99999$, a=0.9868, τ=0.0990, $R_{0}$ = 11,005, $R_{\infty }=1003.6$).

**Table 2 sensors-20-04686-t002:** Extracted Cole–Cole parameters and relative errors (Equations (Equation 20)–(Equation 23)) calculated from 80-point BIS data.

Meas. No.	$R^{2}$ (-)	*a* (-)	$\epsilon_{a}$ (%)	τ (s)	$\epsilon_{\tau }$ (%)	$R_{0}$ (Ω)	$\epsilon_{R_{0}}$ (%)	$R_{\infty }$ (Ω)	$\epsilon_{R_{\infty }}$ (%)
1.	0.99997	0.9784	2.22	0.1005	0.47	11,046	0.42	999.22	0.08
2.	0.99999	0.9876	1.26	0.0992	0.876	10,967	0.30	1003.6	0.36
3.	0.99998	0.9797	2.07	0.10002	0.02	11,033	0.3	1000.2	0.02
4.	0.99997	0.9790	2.14	0.1003	0.35	11,041	0.37	999.44	0.06
5.	0.99997	0.9794	2.10	0.1002	0.21	11,037	0.34	999.87	0.01

**Table 3 sensors-20-04686-t003:** Extracted Cole–Cole parameters and relative errors (Equations (Equation 20)–(Equation 23)) calculated from 264-point BIS data.

Meas. No.	$R^{2}$ (-)	*a* (-)	$\epsilon_{a}$ (%)	τ (s)	$\epsilon_{\tau }$ (%)	$R_{0}$ (Ω)	$\epsilon_{R_{0}}$ (%)	$R_{\infty }$ (Ω)	$\epsilon_{R_{\infty }}$ (%)
1.	0.99999	0.9877	1.25	0.0989	1.10	10,998	0.03	1003.9	0.39
2.	0.99999	0.9870	1.32	0.0992	0.80	11,004	0.04	1003.4	0.34
3.	0.99999	0.9881	1.21	0.0987	1.26	10,993	0.07	1004.3	0.42
4.	0.99999	0.9895	1.06	0.0985	1.52	10,978	0.20	1004.9	0.49
5.	0.99999	0.9868	1.34	0.0990	1.01	11,005	0.05	1003.6	0.36

**Table 4 sensors-20-04686-t004:** Averages of extracted Cole–Cole parameters.

Meas. Type	*a* (-)	τ (s)	$R_{0}$ (Ω)	$R_{\infty }$ (Ω)
40 point	0.9806	0.1001	10,999	1001.5
80 point	0.9808	0.1000	11,025	1000.5
264 point	0.9878	0.0989	10,995	1004.0

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
