# Peer review of "Physical Validation of a Residual Impedance Rejection Method during Ultra-Low Frequency Bio-Impedance Spectral Measurements"

_sensors, 2020, doi:10.3390/s20174686_

Round 1
Reviewer 1 Report
Comments:
- The author points out that according to reference 31, this paper improves the software and optimizes the four channel instrument, so the specific improvement and optimization should be given in detail.
- The performance of the realized complete measuring system proposed in this paper should be proved in detail, such as accuracy, noise, etc.
- The result figuresshow that the measured value is highly consistent with the theoretical calculation. The author should give a more detailed description of the measurement method. In addition, what are the samples and what constraints does the system have on the measured samples and conditions?
- There should be a more detailed introduction to the measurement system, adding the principle description, schematic diagram, physical diagram, etc., for example, in Figure 3, only the appearance photo is not enough, the physical picture of the internal circuit of the system should be added.
- If the measurement of actual samples is not involved, the title should reflect that this paper is a numerical simulation.
- There should be a comparison between the methods and those commonly used at present.
- There are too many figures, unnecessary ones can be deleted, and some can be combined together.
- The tables arenot standardized and should be in the form of three line table with the title at the top.
- The structure of the article is somewhat confusing, for example, there is only 1.1 but no 1.2, and the subheading should no longer be "1.2.3.".
Reviewer 2 Report
The manuscript presents a method for residual impedance rejection from the impedance spectral measurements.
The manuscript is oriented to the biological applications and in fact it represents a continuation of the authors work, previously published, for improving their BIS measurements system through a new proposed method.
The manuscript is well written and it is interesting for the research community, but some aspects that are listed below should be clarified.
The measurement system is described in a general way, without mentioning the used devices (that should be done in order to be able to understand the system performances). What microcontrollers, ADC or DAC were used? The used communication protocol is I2C or I2S? Is the excitation generator based on MCU1 or there is an external device? At line 170 is mentioned that “the excitation signal is a monochromatic sine wave in the frequency range between 1 MHz to 100 kHz”. Is it correct? (I believe that the domain starts from 1mHz, and not from 1MHz –comparing to the previous published paper). At the lines 172-173 is mentioned that the voltage generator has the frequency range of 0.1 Hz to 40 kHz. It is not clear why there are the differences in the frequency ranges of the excitation signal and voltage generator. The authors should clarify these aspects.
I noticed that once with the increasing of the number of points on decades, the errors are increasing. The reason of such behaviour is not discussed by the authors. Some comments on this are welcomed.
In order to strengthen the proposed method, the authors should include some discussions about the impact of the noise in the measurements over the method’s performance.
In the conclusions section, at lines 324-325, is mentioned that “Authors from different fields intend to use the method described above in their own fields of research.” This affirmation looks like a speculation. I recommend to rephrase it, oriented on possible applications.
Round 2
Reviewer 1 Report
I think the revised version of this manuscript has basically solved my doubt. I only have one suggestion. All the subtitles of different parts of a Figure should be placed below the picture.
Such as : Figure 1 bla bla (a)bla bla bla (b)bla bla bla(c)bla bla bla...
Author Response
Dear Reviewer!
Thank you very much and we are pleased with your comment on our manuscript.
Comment:
"All the subtitles of different parts of a Figure should be placed below the picture.
Such as : Figure 1 bla bla (a)bla bla bla (b)bla bla bla(c)bla bla bla..."
Response:
"Thank you for your suggestion. As you can see in the attached manuscript, we have placed the titles of the figure as suggested. In addition, this article was finalized using the MDPI english editing service (English editing ID: English-21331)."
We look forward to seeing our answers as satisfactory.
Sincerely,
Zoltan Vizvari
Department of Environmental Engineering, Faculty of Engineering and Information Technology, University of Pecs,
Boszorkany str. 2, H-7624 Pecs, Hungary
vizvari.zoltan@mik.pte.hu

Reviewer 2 Report
The manuscript is improved.
A small issue should be checked:
Line 288: Is the value for the duration (time interval) of the measurements (100KHz-10Hz) correct?
Author Response
Dear Reviewer!
Thank you very much and we are pleased with your comment on our manuscript. Our response to your comment can be found below!
Comment:
"Line 288: Is the value for the duration (time interval) of the measurements (100KHz-10Hz) correct?"
Response:
"The value in the line 288 is due to the fact that the duration of measurement at that frequency interval (100KHz-10Hz) is 33 s/decade. Because the interval consits of four decades, the total duration of measurement in this interval is four times 33 s/decade (132 sec)."
We look forward to seeing our answers as satisfactory.
Sincerely,
Zoltan Vizvari
Department of Environmental Engineering, Faculty of Engineering and Information Technology, University of Pecs,
Boszorkany str. 2, H-7624 Pecs, Hungary
vizvari.zoltan@mik.pte.hu